# Sustainable Manufacturing Technologies: A Systematic Review of Latest Trends and Themes

**Ali Bastas** 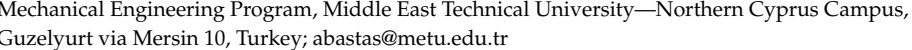

Mechanical Engineering Program, Middle East Technical University—Northern Cyprus Campus, Guzelyurt via Mersin 10, Turkey; abastas@metu.edu.tr

**Abstract:** Meeting current needs while not sacrificing the future ability to do so as a key sustainability concept is becoming more challenging than ever, with the increasing population rate, energy poverty, global warming, and surging demand for products and services. Manufacturing is in a prime position to address this challenge, with its significant economic contribution to the global GDP and its high influence over the environment and humanity. Sustainable manufacturing technologies research is growing to support our journey towards sustainable development. This article undertook the systematic review of state-of-the-art sustainable manufacturing technologies literature, evidencing the latest themes and trends in this important research avenue. Descriptive and thematic analyses were performed, synthesising the latest advancements in the field. Sustainable manufacturing processes, especially sustainable machining, was established as a key theme, including research endeavours of elimination of lubricants. Various manufacturing systems and process sustainability assessment technologies were noted. Sustainability indicators addressed were critically evaluated. As an outcome, a conceptual framework of sustainable manufacturing technology research was constructed to structure the knowledge acquired and to provoke future thinking. Finally, challenges and future directions were provided for both industrial and academic reader base, stimulating growth in this fruitful research stream.

**Keywords:** sustainability; sustainable manufacturing; manufacturing processes; green manufacturing; energy efficient manufacturing; manufacturing systems

## 1. Introduction

Sustainable development is now an imperative parameter, commonly defined as the "development that meets the needs of the present without compromising the ability of future generations to meet their own needs" by the Brundtland Report [1,2]. Elkington [3] converted this key phenomenon into the multidimensional concept of triple-bottom line, holistically framing sustainability as the satisfaction of not only economic agendas but also environmental and social requirements. Given the consistent increase in the world population [4] and the associated rise in demand and consumption rates of products and services [5–7] sustainability science and engineering technology research streams are growing at macro and micro levels with a view to guide governments and organisations towards sustainable development [8–11].

Manufacturing, with its wide scope and broad range of stakeholders, "historically had been and still is the key factor for the development and prosperity of nations" [12]. According to Haraguchi et al. [13], the manufacturing sector has an unaffected, value-added contribution to world GDP and employment since 1970 and will be an engine of growth for the developing countries. Although the recent global growth rate of manufacturing has been negatively influenced by the COVID-19 pandemic, it still remains as an avenue of paramount importance for all countries to meet their 2030 Sustainable Development Goals [14]. Customers are also increasingly demanding products and goods that have been sustainably manufactured [15] and would potentially be willing to pay more for sustainable products [16,17].

With these facts, it can be clearly articulated that manufacturing will remain as a key influencer of economy of nations and firms; social parameters, such as labour practices and Occupational Health and Safety (OH&S); and environmental issues, such as energy consumption, waste and effluents, and emissions [18–20]. Given these clear links to all or most of the indicators of national and organisational sustainability, the concept of sustainable manufacturing (SM) emerged and is exponentially growing with a view to enable a response to "the sustainability challenge" that we are facing, through innovative systems, models, processes, and technologies [19–23].

Furthermore, "sustainable industrialisation through promoting new technologies" was outlined as one of the 17 Sustainable Development Goals [24–26], as part of the UN's 2030 sustainability vision, strategically emphasising the importance of development of sustainable manufacturing technologies. The manufacturing technologies are a "vital ingredient" of engineering, consuming energy, and using manpower as required in a systematic manner with a view to convert raw materials and resources into products useful for society [18,27,28]. Stark et al. [29] outlined the scope of manufacturing technologies as "the development of production technologies, machine-tool concepts and factory techniques" formulating the processes required "to ensure whatever has to be produced, it can be done with economy of resources which likewise uphold social standards." It was established that the manufacturing processes have a clear impact on the sustainability impact of firms [30]; however, innovation and technological progress through research are key to establishing sustainable manufacturing and industrialisation solutions [14,31,32]. This was resonated by Lopes de Sousa Jabbour et al. [33] and Shi and Li [20], who concluded innovation and technology's remarkable impact on sustainable development of manufacturing firms.

Several reviews have been undertaken to date on the emerging research avenue of sustainable manufacturing. Jayal et al. [34] reviewed the sustainability assessment and optimisation models at the product, process, and system levels for sustainable manufacturing, presenting a case study on the machining processes. Despeisse et al. [21] established the types of sustainable manufacturing activities, outlining the best practices for industry. Rosen and Kishawy [23] stressed the importance of integrating sustainability into design and manufacturing, deducing that further and more comprehensive research studies are required to support such an integration through technology transfer and applications. Research trends and challenges were analysed, and key sustainable manufacturing research clusters were established as "business models and processes, asset and product life cycle management, resources and energy management and enabling technologies" [19]. The indicators of SM were categorised [35]. Haapala et al. [18] reviewed engineering research in SM, revealing clear challenges in manufacturing process and system research, development, implementation, and education. A comprehensive content analysis was undertaken into the concept of SM and its definitions [36]. Gbededo et al. [37] captured the SM approaches between 2006 and 2018 and initiated a simulation and life-cycle analysis based manufacturing system sustainability assessment tool. More general and rather abstract reviews of research trends in SM were more recently conducted by Lee et al. [22] and Yoon et al. [38]. The synergistic link between SM and Industry 4.0 was systematically reviewed [39,40].

It can be seen from the extant literature reviews on SM that further developments and innovations are highly required [22,41], and manufacturing technologies form the foundation of realising SM [18,19,32,41]. On the other hand, although it is of clear importance in enabling the sustainability transition of manufacturing, structured, specific, and comprehensive research from the lens of sustainable manufacturing technologies and engineering remains highly limited or has not been carried out recently [41]. Despite findings by Lee et al. [22] and Yoon et al. [38] in the area, a particular manufacturing technology and engineering focus, along with a rigorous analysis of the latest research in this highly emerging and fruitful avenue, would be further valuable for both academics and practitioners. Further, as articulated by Kuhn [42], "there is no one objectively correct understanding of earlier research and knowledge contributions," and diversity of views and outcomes stimulate debate and advancement of the associated field [43]. Stemming

from these standpoints, this study carried out a detailed systematic review towards identifying the state-of-the-art research in SM technologies, as schematically represented in Figure 1, with a view to address the following research questions:

- What are the latest trends and themes in the sustainable manufacturing technology research?
- Which SM technology research areas are recently receiving attention by the literature?
- Which dimensions of sustainability (triple-bottom line) are being addressed by the recent SM technology research?
- What are the sustainability indicators being analysed by the SM technology research?
- What are the challenges, requirements, and directions for future SM technology research?

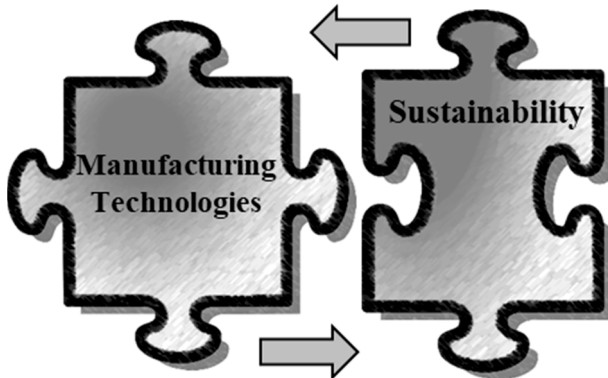

**Figure 1.** The aim and scope of the literature review.

The main motivation behind this research was to generate a current state map of the sustainable manufacturing technologies research, with a view to synthesise key themes, gaps, and opportunities; outline future research directions; and foster growth in this important research avenue.

The subsequent sections of this paper contain the following: Section 2 comprehensively discusses the research materials and methodology adopted in this systematic literature review; the descriptive findings of the study and key themes identified are presented in Section 3; a conceptual framework contribution of sustainable manufacturing technology is introduced in Section 4; the implications of the review along with discussions on the challenges and future directions are provided in Section 5; and finally, conclusions are outlined in Section 6.

## 2. Materials and Methods

Systematic literature review (SLR) is a widely adopted method in sustainability research [44–46]; sustainable manufacturing [22,37]; technology engineering [47]; and various other engineering disciplines, including mechanical engineering [48,49], software engineering [50], and other engineering fields [51,52]. Such an establishment of systematic literature reviews is mainly due to its objective and evidence-based nature, facilitation of analysing diverse knowledge bases, its methodological rigour, its transparency, identification of "known" and "unknowns" in the associated avenue of inquiry, and acting as an engine of growth [43,44,53–56].

There are five fundamental phases of systematic literature reviews, which were deployed in this study, as demonstrated in Figure 2. The phase 1 included the formulation of the research strategy and the identification of the research questions, as discussed in Section 1; the materials, including the associated journals and databases, were located and established in phase 2; the captured materials were then reviewed for inclusion and categorised in phase 3; descriptive and thematic analyses of the materials were undertaken in phase 4; and findings were summarised and reported in phase 5 [43,53–55].

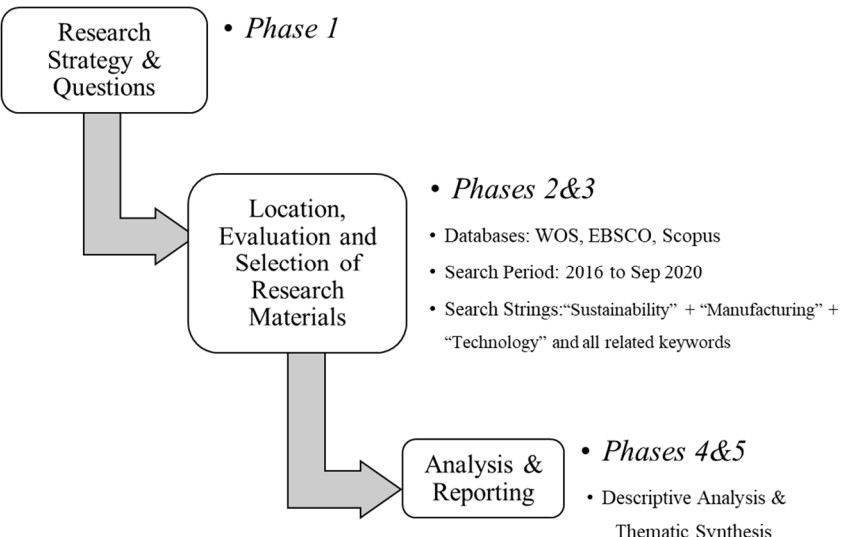

**Figure 2.** Systematic literature review (SLR) phases deployed in the study.

Journal and conference publications, as per the research questions and scope, were identified using the three key aggregator databases highly relevant to sustainability, manufacturing technology, and engineering areas, including EBSCO (ebscohost.com), ISI Web of Science (wokinfo.com), and Scopus (scopus.com). Despite the adoption of three major aggregate databases leading to overlap of information, it ensured capturing of key information in the literature, and duplicates were managed and removed using Elsevier's Mendeley software [37]. The search was limited to peer-reviewed publications in English language, with a view to ensure reliability and rigour of the studies included [55,57]. The search period for the study was set as last 5 years (2016 to Sep. 2020) to ensure focus on the recent materials in the field, enhance rigour of the analysis, and to enable the state-of-the-art nature of the systematic review analysis conducted. Producing unpublished data, shedding light onto grey areas in the literature, and focusing on analyses that have not yet been carried out, in line with the research objectives, should form the primary objective of systematic literature reviews, and researchers should feel encouraged to narrow the scope (time period, etc.), as required, to ensure rigour [43,53,55]. Such a 5-year focus and delimitation period was also adopted by Lee et al. [22], i.e., 2013 to 2017, stemming from a similar rationale. Yoon et al. [38] covered SM works until 2015.

The inclusion criterion was set as published works that introduced SM technology engineering advancements, including the manufacturing process and system or process design and cloud technologies, providing integration and improvements in any one or all of the triple-bottom line dimensions of sustainability (i.e., environmental, economic, and social). Studies outside the sustainable manufacturing technology and engineering context, such as organisational implementation and management aspects of SM, sustainable business models, and sustainable supply chain management, were categorized as irrelevant and were excluded from this review, as there were recent and extensive reviews present on these areas, such as Centobelli et al. [58] and Mardani et al. [59]. Moreover, research papers that were identified within the manufacturing technologies context in the absence of any emphasis or links to sustainability dimensions were omitted, in line with the established sustainability scope and agenda of this research. Search strings of "sustainability," "manufacturing," and "technology," and their related keywords, e.g., "sustainable machining; sustainable manufacturing system; sustainable manufacturing process; sustainable casting; and etc." were adopted in the aforementioned databases, identifying a significant number of published works discussing a broad range of issues.

As a result, the SLR protocol adopted captured a wide range of issues within the sustainable manufacturing technology research domain, not limited to but including SM processes, e.g., machining, additive manufacturing, digitalization technologies for SM such

as Industry 4.0 and Maintenance 4.0, energy and efficiency improvement technologies, cleaner production instrumentation, performance evaluation techniques, and optimization models. Reference lists of reviews conducted in the area and other indicative work were also checked and compared with the outcomes of the searches conducted, and the robustness of the search was verified [53].

Descriptive statistics were utilized as one of the primary analysis methods with the aid of a MS Excel database, where the key descriptive information, such as the publication year, country of the corresponding author, etc., regarding the published works confirmed for inclusion were captured, analysed, and reported [44]. Moreover, for a robust and systematic synthesis of the key themes in the literature, a thematic synthesis method was selected [60–62]. The key themes and relevant information in the included works were documented, classified, disassembled into codes and then reassembled into common themes according to these codes, results interpreted critically, and conclusions were drawn [60–62].

## 3. Results

Following the outlined systematic literature review protocol, the articles identified were captured, an initial eligibility check carried out through review of title and abstracts, and confirmed for inclusion post-review of full texts in the review through an iterative selection process, as demonstrated in Figure 3 [44,63].

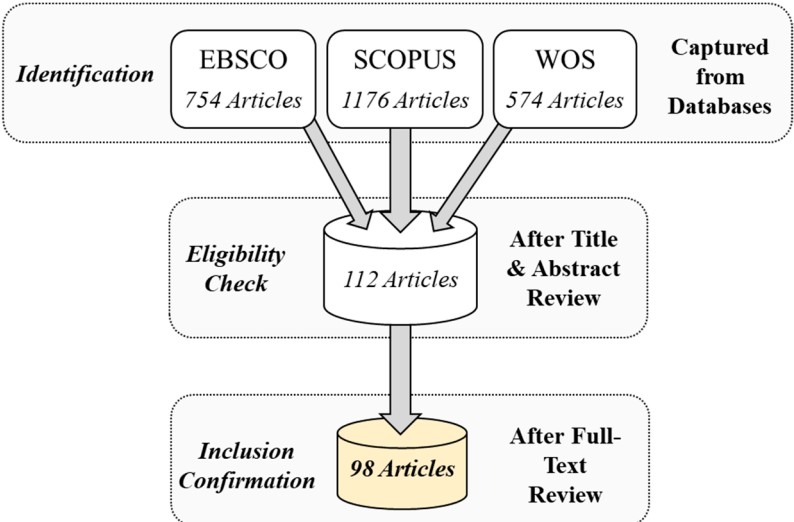

**Figure 3.** Article identification, review and inclusion process.

The selection and inclusion was undertaken strictly in line with the research questions and inclusion/exclusion criteria established as part of the SLR process, and, as a result, 98 articles relevant to the study were identified and carried forward to descriptive and thematic analyses stages [63].

### 3.1. Descriptive Analysis

The distribution of articles included against publication years are shown in Figure 4, where the emerging and growing nature of the sustainable manufacturing technology research is further evidenced. This observation resonates with and complements the extant review studies conducted on SM that documented the research focus in the area in the past 10 years [19,22,34,37,38]. Furthermore, 23% of the materials included in the review were recorded in 2020 (until September), demonstrating the highly recent focus in the area.

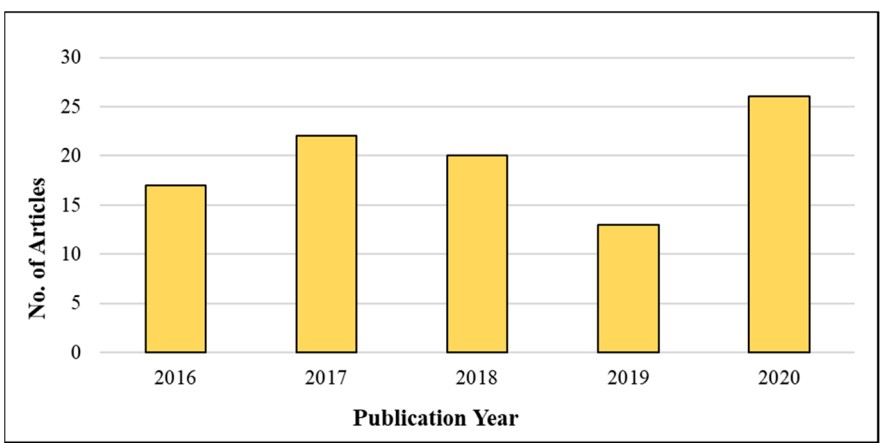

**Figure 4.** Number of publications per year.

Taking into account the sustainable development challenges that our society is facing and the UN vision [14,25], further growth and focus is envisaged in the SM technology area, as SM is a "journey" and not a "destination" [33,64].

The geographical location of the corresponding authors of published works included was extracted, and findings are presented in Figure 5. The Chinese authors' prominence in the SM technology research was noted, with a major portion (22%) of all works conducted in China. This can be an outcome of the major funding support to research and development (R&D) activities in this region [65], along with a solid focus on manufacturing technologies development [66]. India followed China with a significant level (20%), and the key role of these countries in SM innovations was further documented, given their important position in the world markets as leading manufacturing nations [67].

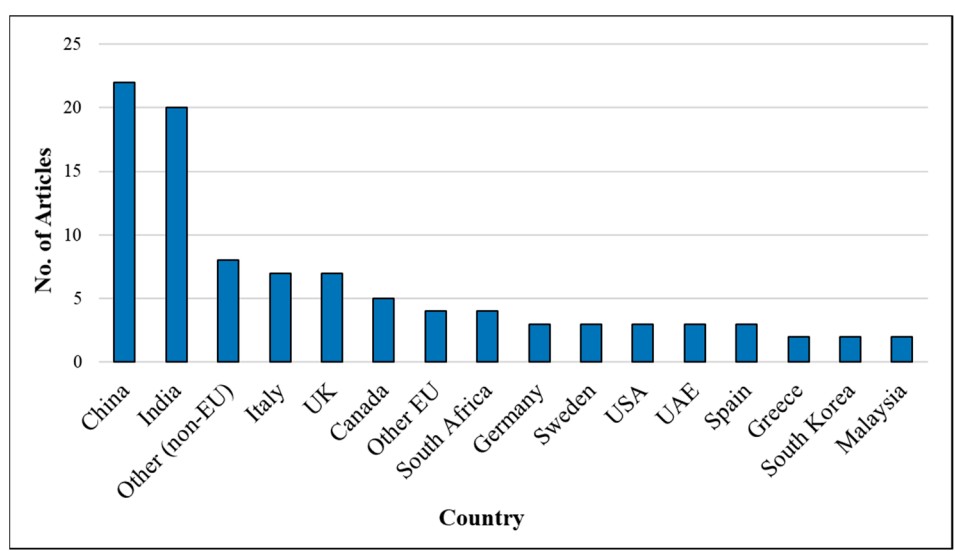

**Figure 5.** Number of publications against geographical region of the corresponding author.

On the other hand, 30% of the publications were found to be compiled in the European Union, including Germany, Sweden, and Spain (with 3% each). Italy and the UK were observed to make a noteworthy contribution (with 7% each). Authors from 27 different countries were identified, in total, from the five continents of Asia, Europe, North America, South America, and Africa, evidencing the "global" research motivation towards SM technologies [68]. Additionally, 38 distinct journals from a range of publishers were identified with various contributions, documenting both the diverse nature of the SLR study undertaken and the wide base of attention that the SM technology research is receiving.

The distribution of the SM technology literature from the research methods point of view is shown in Figure 6. Research studies of explanatory nature, accommodating statistical and experimental approaches, were observed to be the main method in state-of-the-art SM technology literature, with 50% of published works adopting this method. This finding echoes with authors such as Creswell [69], Wieringa and Heerkens [70], and Petersen and Gencel [71], evidencing the engineering and technology researchers' significant tendency towards a positivist worldview and objective research methods, although there are certain advantages in adopting qualitative approaches in engineering design research [72]. Case studies were also a common research method (32%), which were mainly quantitative and were utilised to demonstrate the application of the new technologies developed [73].

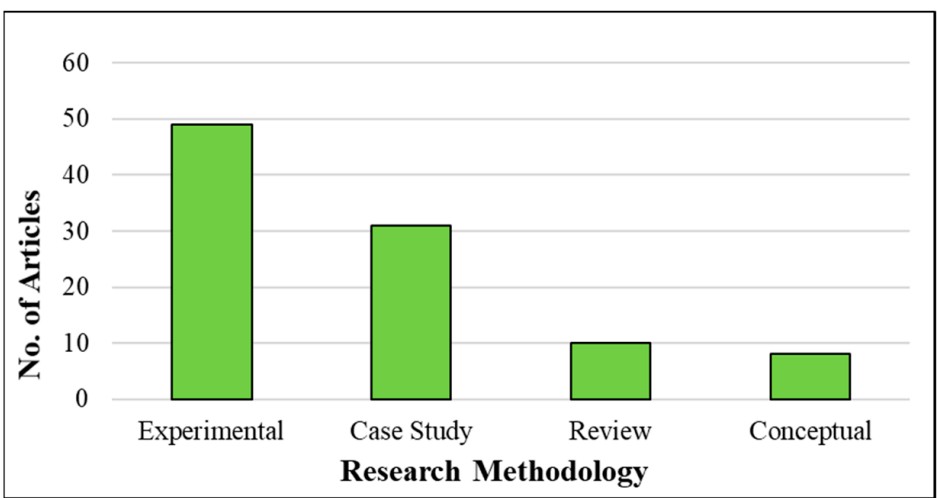

**Figure 6.** Number of articles against research methodology applied.

The articles were categorised into the sustainable manufacturing technology area studied, and findings are demonstrated in Figure 7. It was noteworthy that a major portion (approximately 50%) of the SM technology research focalised in sustainable machining technologies [74,75], including Alvarez et al. [76], Uhlmann et al. [77], and Jawahir et al. [78], who documented various sustainable machining and engineering solutions. Welding is an impactful manufacturing process for sustainability [79,80], and it was identified as the second most popular SM technology research area, with 14%. Additive manufacturing as a highly emerging research stream [81] was established as the third key theme (13%), with various sustainability aspects of this promising manufacturing technology receiving attention. Various manufacturing process design and optimisation approaches for sustainability improvement were also identified as significant (10%). These key themes were further analysed and complemented through thematic synthesis in Section 3.2.

The sustainability dimension(s) addressed by the published works were identified, as presented in Figure 8. Only 21% of the articles addressed or incorporated all three dimensions of sustainability through the holistic lens of triple-bottom line (TBL). On the other hand, environmental or green sustainability was the key focus area of SM technology research, with 74% of the articles studying the environmental sustainability dimension either in isolation (25%) or together with the economic or social sustainability considerations (e.g., energy consumption, waste, and/or carbon emission issues). Such a focalisation against the environmental dimension is not uncommon in sustainability integration research streams [44,82,83]. This trend is mainly due to the recent and highly alarming global warming [84] and energy sustainability challenges [85] that our society is facing.

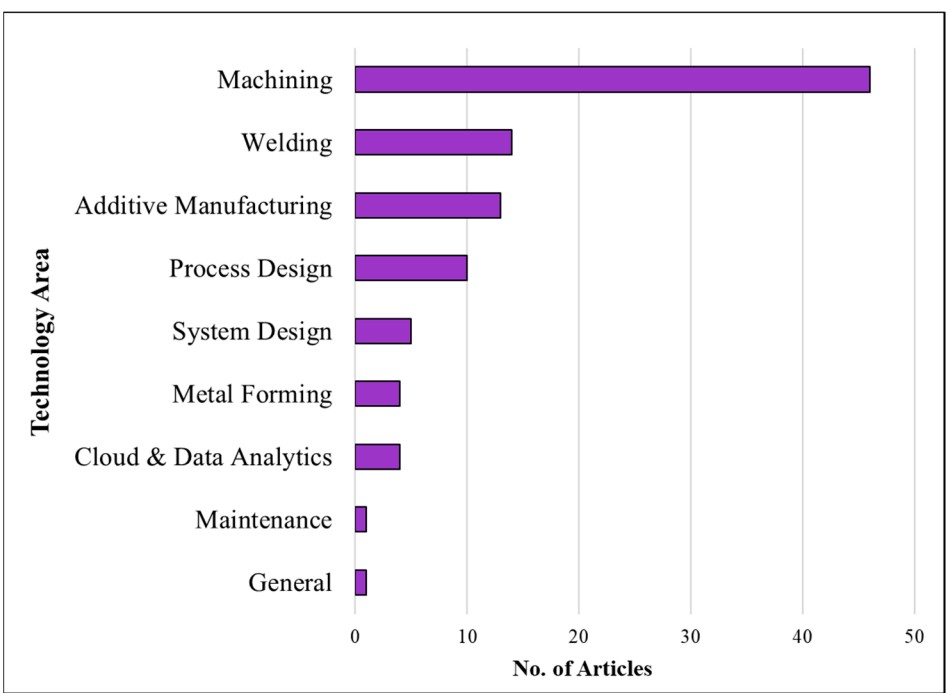

**Figure 7.** Number of articles against the sustainable manufacturing (SM) technology area studied.

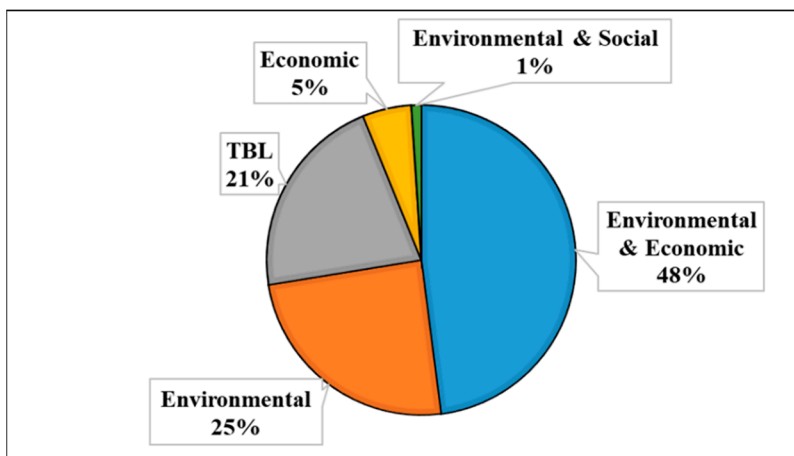

**Figure 8.** Distribution of papers against the sustainability dimensions.

Moreover, the manufacturing industries and construction accounted for around 20% of global carbon emissions in 2014 [86], and manufacturing consumes a significant portion of energy in most countries, e.g., 50% of total electricity produced in Germany [87]; therefore, improving the environmental sustainability performance of manufacturing processes, including machining, is of paramount importance [77]. The social dimension was mainly captured by the studies that adopted the complete TBL view; however, the social benefits of the SM technologies developed, such as reduction/elimination of hazardous substances, including lubricants for enhanced operator safety, was an indirect benefit of the environmental sustainability improvements introduced, and yet, there was no observable or specific mention of this dimension. The particular sustainability indicators studied by the SM technology works were analysed further, established, and synthesised in Section 3.2.3.

The workpiece materials adopted in the experimental and case study research studies are shown in Figures 9 and 10. The titanium alloy of Ti-6Al-4V was noted as the most popular material utilised in the latest SM technology research (18% of studies utilising workpiece-based data), due to its superior material properties, e.g., high strength, low

density, high fracture toughness, corrosion resistance, and biocompatibility [88]; high industrial demand, including its well-established history in the aerospace sector [89]; and its suitability to various manufacturing technologies, including additive manufacturing [90].

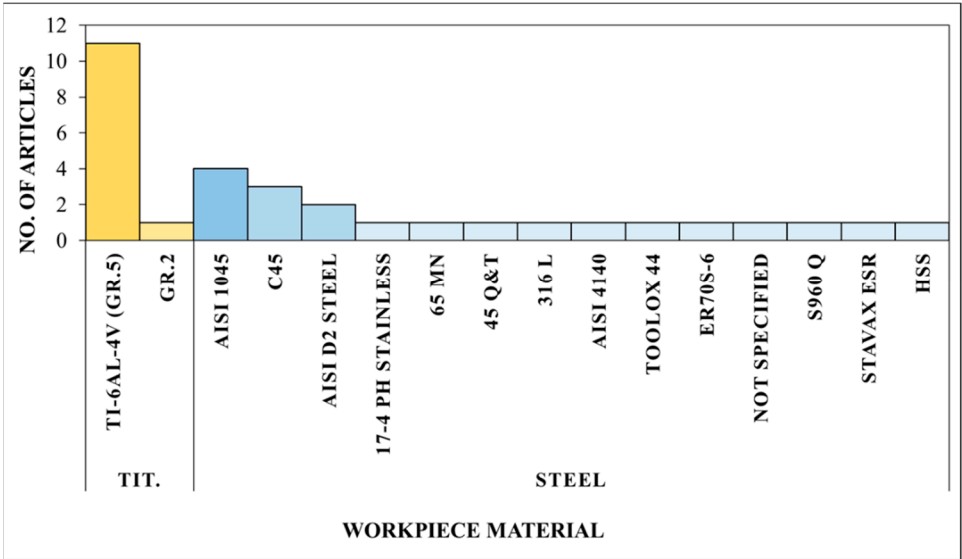

**Figure 9.** Distribution of papers against the workpiece materials (titanium and steel).

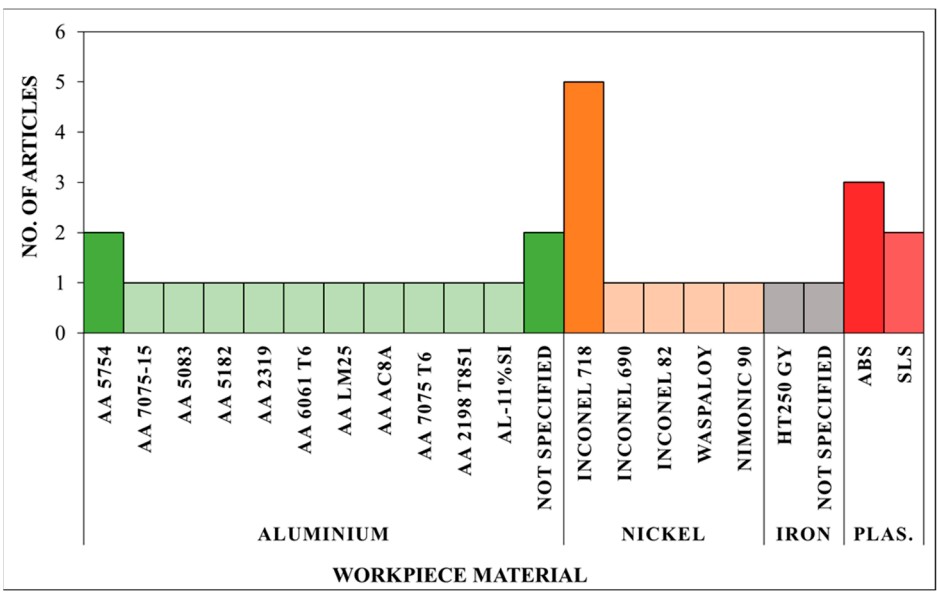

**Figure 10.** Distribution of papers against the workpiece materials (aluminium, nickel, iron, and plastics).

On the other hand, steel was observed to be the most popular workpiece material (32% of studies utilising workpiece-based data), with SM technology research studying various grades of this highly important engineering material [91], including the AISI 1045 [92,93] and C45 [94,95] grades.

Aluminium alloys, such as AA 5754 [96] and AA 7075 T6 [97], nickel alloys [98], and cast iron [99] also received noteworthy attention from the SM technology research. Among the nickel alloys, Inconel 718 was observed to be a popular workpiece material (8% of studies utilising workpiece-based data), which is a high performance super alloy that offers corrosion resistance along with strength at both atmospheric and high temperature ranges [100]. Further, plastic materials, such as ABS and SLS were studied in Additive

Manufacturing technology research for sustainability improvements, such as performance analysis of manufacturing through reclaimed plastic powders [101,102]. All in all, 37 different workpiece material grades were identified from six base materials (i.e., titanium, steel, aluminium, nickel, iron, and plastics), outlining the diversity of engineering materials adopted as workpieces in SM technology research.

### 3.2. Thematic Synthesis and Analysis

3.2.1. Thematic Map of Sustainable Manufacturing Technology Research

The focal research streams and themes surrounding the SM technology research are presented in Figure 11, along with weightings of recurrence (percentage of papers addressing the identified themes). Manufacturing processes are of paramount importance for manufacturing organisations [103] and were identified at the heart of sustainable manufacturing technology research, with 87%.

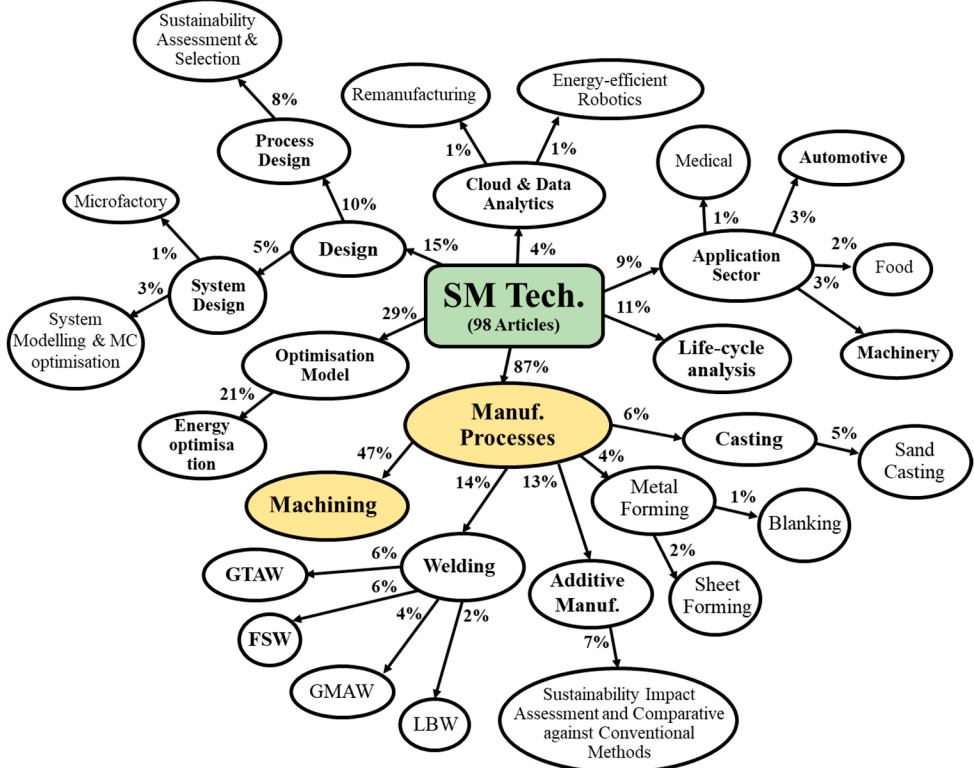

**Figure 11.** Concept map of the sustainable manufacturing technology research, demonstrating various research streams identified and their distributions.

Although various process technologies, such as additive manufacturing [104–106]; laser manufacturing, which utilises the materials from additive manufacturing [107]; welding [108]; casting [99,109]; and metal forming [110,111] received attention, and machining technology research was established as the main theme, with 47%. This observation further highlights machining's significance among manufacturing processes [112], its sustainability constituting a key research avenue [74,75,77]. Sustainable machining as a key theme was analysed in further detail in Section 3.2.2.

Various sustainability aspects of additive manufacturing (AM) was studied, including the sustainability impact assessments of this emerging technology [113–115]; comparative studies against conventional manufacturing techniques [116,117]; and use of more sustainable additive manufacturing techniques [118] and materials, including performance evaluation of recycled materials [101].

Welding, as an influential process for various manufacturing sectors [27], received significant attention. Sustainability assessment, multicriteria decision-making (MCDM),

and welding process parameter optimisation studies for sustainability improvements were carried out for friction stir welding (FSW) [96,119,120], gas tungsten arc welding (GTAW) [121–123], gas metal arc welding (GMAW) [79,124], and laser beam welding (LBW) [108,125].

Quantitative modelling and optimisation studies were a focal area (with 29%), with optimisation models at both system [126,127] and process levels [128–131] being developed for offering sustainability improvements, in particular, for energy efficiency improvements. Manufacturing system and process design technologies for assessing, selecting, and providing basis for more sustainable designs were documented [132,133].

Life-cycle assessment and analysis (LCA) was also established as a popular concept in SM technology research [134,135], with LCA being adopted to assess life-cycle sustainability impacts of proposed SM technologies [133], such as the laser AM technology [136], welding technologies [79], and lubrication technologies [137].

Several studies investigated their proposed SM technologies through application case studies. Automotive [109,138,139], machinery [133,136,140], food [113,127], and medical [114] sectors were noted as the application sectors in the latest SM technology research.

### 3.2.2. Thematic Map of Sustainable Machining Technology Research

As the sustainable machining literature represented approximately the half (50%) of the SM technology research, an additional thematic synthesis and further analysis was undertaken on this focal area, the results of which are presented in Figure 12.

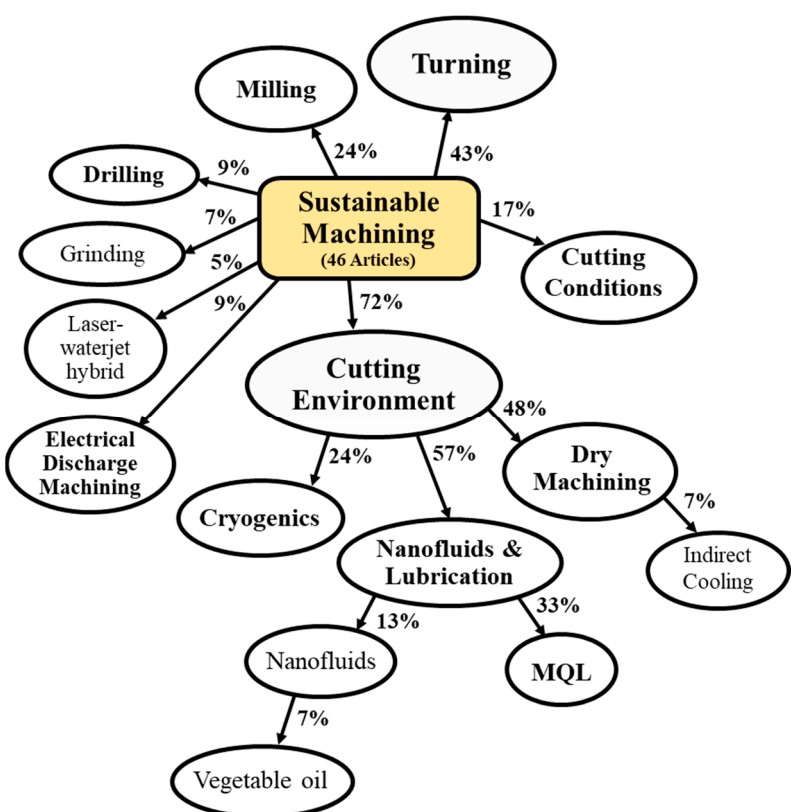

**Figure 12.** Concept map of the sustainable machining technology research.

Cutting environment was evidenced as the key theme of the latest sustainable machining body of knowledge (70% of articles categorised under sustainable machining), with SM technology studies studying the elimination of lubrication substances through cryogenic [138,139], dry machining [128,140] and indirect cooling [141], or their reduction/optimisation through minimum quantity lubrication (MQL) techniques [142,143].

A current endeavour present in the sustainable machining research is elimination or reduction of lubrication substances, which is highly impactful for economic (reduction of costs), environmental (reduction of hazardous substance control and waste treatment), and social (reduction of health and safety risks to workforce) sustainability improvements. Yip and To [144] resonated with this viewpoint, articulating environmental issues, especially regarding machining of difficult-to-cut materials that have high use rates of lubricants, and associated waste disposal challenges and concerning pollution rates. Nanofluids were established as an emerging research avenue, with a number of nanofluids, such as $Al_2O_3$-MWCNT being proposed as lubricants for sustainability improvements [145–147]. Vegetable oil-based nanofluids were further noted as a highly emerging advancement in the sustainable nanofluids and lubrication technology research [148–150].

Turning (43% of articles categorised under sustainable machining) and milling (with 24%) were noted as the two most popular machining processes in the SM technology research. On the other hand, sustainable machining literature further studied a number of other machining processes, including drilling [92,151], electrical discharge machining [152,153], grinding [154], and laser–waterjet hybrid machining [155].

Moreover, cutting conditions improvement and optimisation studies were evident (with 17%) [156–158], which investigated optimal parameters, including material removal rates; depth of cut; feeds and speeds for sustainability improvements, including optimising energy consumption [153]; and reducing carbon emissions [152]. Nur et al. [159] investigated various cutting conditions (i.e., cutting force), realising energy efficiency improvements for sustainable machining of an aluminium alloy (Al-11%Si).

### 3.2.3. Sustainability Indicators Studied

The sustainability indicators studied in the articles were codified and extracted, and key indicators were identified for each TBL sustainability dimension, as presented in Figure 13. Starting with the social dimension, "health and safety" was established as the key indicator (with 23 articles), with SM technology research studying or implying the effect of SM technologies on the operators, mainly through control of manufacturing substances hazardous to health, such as lubricants [160], air quality [158], and reduction of noise pollution [123]. "Working conditions" and "employee satisfaction" were further observed as other significant indicators of social sustainability, in the context of SM.

From the environmental perspective, "energy consumption" was by far the dominating parameter (with 75 articles), with research streams endeavouring to model [94], optimise [157], and develop the energy efficiency [95] of manufacturing processes for SM. Reduction, elimination, or recycling of manufacturing waste; reduction of $CO_2$ emissions; and reducing resource and material consumption rates were the other environmental sustainability indicators that received significant attention in the SM literature.

Manufacturing cost was the imperative indicator of economic sustainability (mentioned in 37 articles), followed by manufactured product quality (typically surface quality for machining technologies). Tool life was established as another key economic indicator for SM process technology research [144,161], along with manufacturing performance, which entails "tool wear, surface integrity, and chip morphology" for sustainable machining, according to Bordin et al. [140]. Processing time was noted as a significant financial parameter, which was also referred to as processing speed or material removal rate by the machining processes. Manufacturing system or process efficiency and process flexibility (i.e., quick changeovers or rapid switch to manufacturing different products) were also established as noteworthy financial sustainability indicators, in the context of SM.

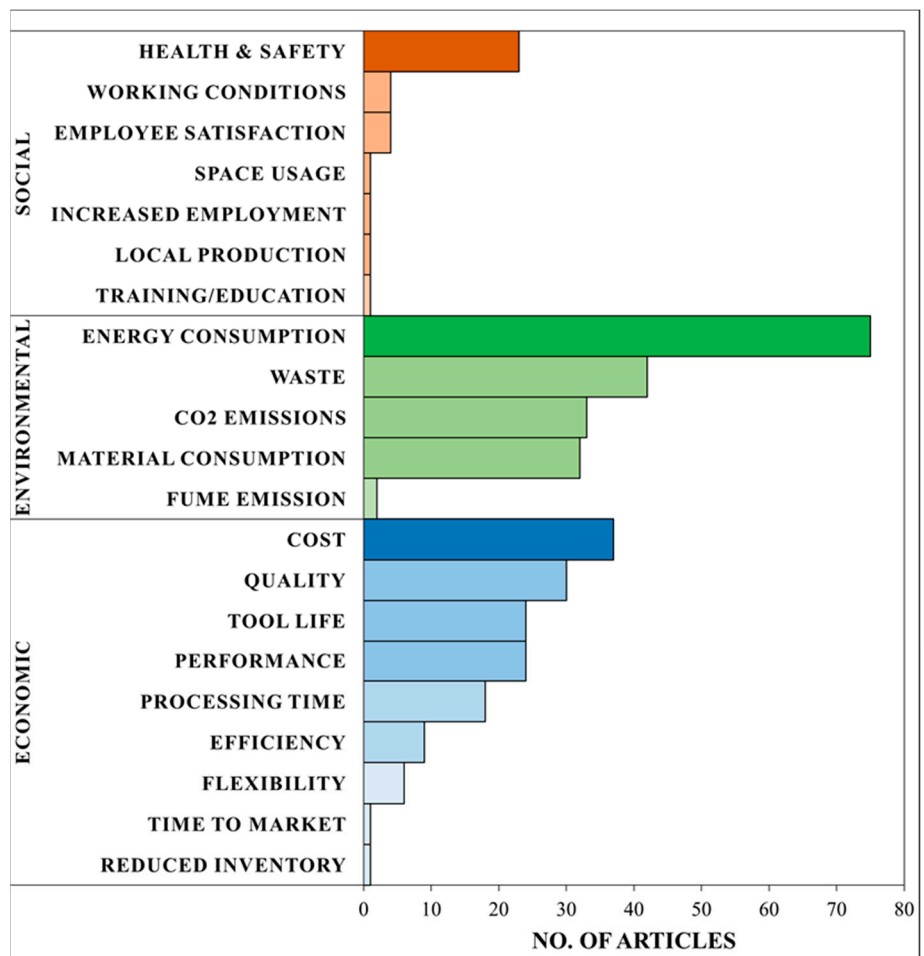

**Figure 13.** Sustainability dimensions and the associated indicators studied by the papers.

## 4. Conceptualisation & Discussion

Conceptual frameworks are constructed through integrating several concepts fundamental to the research phenomenon being studied (i.e., SM technologies in the context of this research), facilitating description or explanation of the associated phenomenon with a view to stimulate holistic understanding and present a map of "the bigger picture" regarding the phenomenon [162,163]. Moreover, conceptual frameworks enable "structuring of knowledge" in engineering research [70]. Stemming from the knowledge accumulation sought from the literature review, a conceptual framework has been constructed, as shown in Figure 14, synthesising the learnings, capturing the research outcomes, and visually framing the state-of-the-art sustainable manufacturing technology research.

Manufacturing system and process design technologies were placed at the foundation of sustainable manufacturing, enabling holistic analyses for designing sustainable manufacturing systems and processes that will proactively support constructing manufacturing entities that are "sustainable-first-time" and contribute to our transition towards SM. Various system and process optimisation techniques [164], tools [109], selection methodologies [133], roadmaps [101], CAD methodologies [165], and models [166] to support this have been offered to date.

Manufacturing process technology innovations, such as sustainable machining [77,78], sustainable lubrication and nanofluids [148], sustainable additive manufacturing [117], sustainable welding [120], sustainable casting [129], and sustainable metal forming [167], are also of paramount importance for SM and will act as "the nuts and bolts" in our journey towards SM.

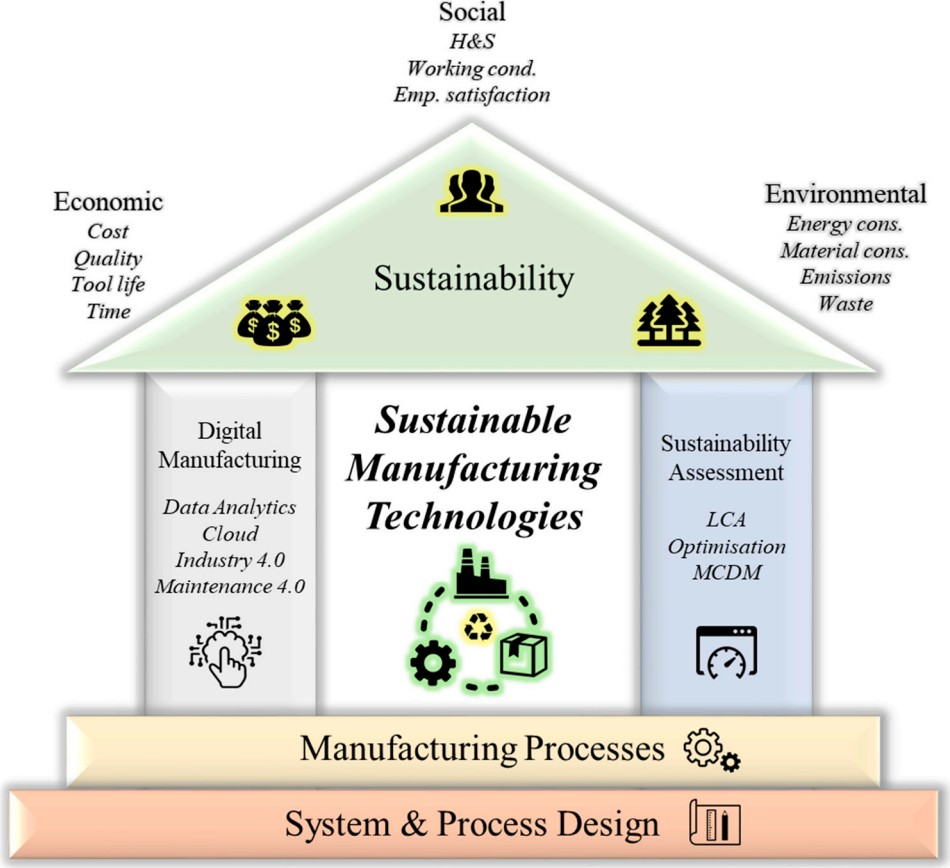

**Figure 14.** Conceptual framework of Sustainable Manufacturing Technology Research.

The digital manufacturing technologies, such as data analytics architectures [168], cloud manufacturing [169,170], intelligent manufacturing systems and metaheuristics [171,172], and Maintenance 4.0 [173] will not only catalyse implementation and dissemination of SM process technologies but also provide meaningful contributions to various SM aspects, including facilitation of remanufacturing [174], optimisation of manufacturing process parameters for sustainability improvements [172], and robotic automation [175].

Ultimately, sustainability assessment is another key pillar of SM [124], which will provide the means for sustainability impact monitoring, decision-making, and continuous improvement through life-cycle assessment (LCA) [136]; multicriteria process parameter analysis and decision-making (MCDM) techniques [176]; and various algorithm-based optimisation approaches [177].

## 5. Challenges & Future Directions

The following challenges were noted for the sustainable manufacturing technology research, along with the associated future research directions:

- **Holistic approach to triple-bottom line sustainability:** It was evident from the findings of this review that there is a major dominance of one dimension of sustainability (i.e., environmental) over the other dimensions (i.e., economic and social). Although there are cases present where the sustainability dimensions and their indicators may complement each other (e.g., reduction of waste not only offers environmental benefits but also social benefits [178,179]), there will be cases where improvement of one dimension may deteriorate performance of the other dimension. Only a limited portion of the literature (evidenced as 21% in this study) was noted to adopt the collective and holistic lens of a triple-bottom line to sustainable manufacturing. Stemming from the principle that "true" sustainability is a multidimensional phenomenon [3,44], the sustainability improvements declared or proposed for specific

dimensions (e.g., environmental) by various contributions in the current body of knowledge are questionable. This is a major challenge that needs to be addressed by future SM technology research holistically evaluating sustainability improvements proposed by the relevant SM technologies and more carefully considering the impacts of their novel contributions. This recommendation is also valid for manufacturing system and process modelling and assessment techniques, which should consider all or most of the sustainability indicators in their assessments and optimisation endeavours.

- **Interdisciplinary engagement:** Manufacturing systems and processes are complex in nature, involving many entities and stakeholders, including human, natural resources, supply chains, and beyond. Hence, there are many relationships and interactions present between this "complex" system and the multidimensional aspects of sustainability, some of which are yet to be revealed and to be understood or to have further insights realised. SM technology research, as further evidenced by this study, has historically been conducted by engineers of technical backgrounds. However, more interdisciplinary cooperation and multidisciplinary efforts towards a real and holistic understanding of the two complex phenomena of sustainability and manufacturing will be catalysed, and future SM technologies aligned, to guide us in our journey towards sustainable manufacturing.

- **Transdisciplinary (industrial) engagement:** It was noted that most of the latest SM technology research was of experimental and academic nature. In addition to interdisciplinary engagement, more transdisciplinary and industrial engagement is particularly encouraged for future research studies, which will not only aid in verification, validation, and dissemination of the proposed SM technologies but also will aid in development of SM technologies that respond to the hot industrial issues, which will positively influence the manufacturing sector's contribution to global sustainable development at an accelerated pace. Industrial collaborations are further envisaged to aid future SM technology research through accommodation of industrial know-how, skills, and resources.

- **Sectoral dissemination:** Only a few studies were observed to apply or validate their proposed SM technologies at a limited number of manufacturing sectors, such as automotive and machinery. Future research is recommended to accommodate more manufacturing sectors, which will not only offer new learnings and knowledge but also facilitate dissemination and familiarisation with future SM technologies.

- **Expansion of scope:** Although the scope of SM technology research was established as significant, limited research was noted on key technologies, such as cloud manufacturing and data analytics; manufacturing system and process design for sustainability; intelligent manufacturing systems; and some fundamental processes, such as grinding and drilling. In particular, state-of-the-art contributions, such as Fountas and Vaxevanidis (2021) highlight the high potential and key role of metaheuristics in driving sustainable development of the manufacturing processes, including the machining processes, facilitating optimisation of process parameters [171,172]. Future SM technology research is particularly encouraged in these fundamental manufacturing areas, which will pave the way for sustainable manufacturing.

## 6. Conclusions

In this paper, a systematic and comprehensive review of the sustainable manufacturing technologies literature was undertaken in line with the research motivation and systematic literature review protocol. A total of 98 papers were identified as relevant to this review between 2015 and Sep. 2020. All research questions were addressed through presentation of detailed descriptive and thematic findings, establishing the latest themes and trends, the research hot spots, and the sustainability dimensions and their associated indicators evident in the SM technology research. The prominent research streams lying at the heart of SM technology were framed through a visual conceptual framework, structuring

the knowledge accumulated over this fruitful phenomenon. Finally, the challenges and directions for future SM technology research streams were outlined.

Peer-reviewed articles in the English language from main databases identified as core to manufacturing, engineering, and sustainability literature were considered in this review, which may have limited the number of articles included and scope of this investigation to a certain extent. However, these measures were taken to ensure the quality of the publications included in the review, and the large sample size of publications considered (98 articles) from a diverse pool of journals (38 journals) offered a holistic view and a high level of reliability for the findings. Moreover, every effort was made in capturing all keywords fundamental to sustainable manufacturing (SM) technology research as per the research objectives, contributing towards the construction of a "complete" view of SM technology.

In conclusion, a resonating focus on sustainable manufacturing processes, especially on sustainable machining, was observed in the state-of-the-art literature. Manufacturing system and process design, modelling, and optimisation studies were established as another prominent research avenue, along with life-cycle analysis of various manufacturing processes. Lubricants for sustainable machining was noted as a research hot spot, with SM technology research seeking to achieve sustainability improvements through elimination or reduction of these typically hazardous substances.

The sustainable manufacturing technology research framework encapsulated system and process design technologies for proactive and sustainable factory designs reinforced by sustainable manufacturing processes and supported by digital manufacturing approaches, such as cloud manufacturing and Industry 4.0. Technologies that enable holistic sustainability assessment of manufacturing systems and processes will provide the basis for continual improvement. Ultimately, future research is recommended that will endeavour to realise holistic sustainability improvements, conscious of the triple-bottom line concept. Interdisciplinary and transdisciplinary research activities will play a key role in addressing the challenges associated with the multidisciplinary and complex nature of driving us towards the "true" sustainable manufacturing, which is a journey and not a destination [180].

**Funding:** This research received no external funding.

**Institutional Review Board Statement:** Not applicable.

**Informed Consent Statement:** Not applicable.

**Data Availability Statement:** Not applicable.

**Conflicts of Interest:** The author declares no conflict of interest.

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
