# Peer review of "Sustainable Manufacturing Technologies: A Systematic Review of Latest Trends and Themes"

_sustainability, doi:10.3390/su13084271_

Round 1

Reviewer 1 Report

Dear Author,

This is a very well structured and organised review paper.

I am proposing a few more references which I believe will enhance your precious work. 

Reviewer 2 Report

Dear Author/s,

I have read with interest your paper and I think it constitutes a very good summary of the state of the art in the topic. 

I think it would benefit from a few changes to better fit with the purpose of the journal, namely, clarifying the use of technical terms and acronyms, for those not familiar with manufacturing processes.

You also state that you are going to provide some conceptual framework, but it doesn't become apparent. In your findings you make a very interesting point, when mentioning that in some works, the improvement in one dimension of sustainability is achieved creating negative impacts in another. This per se would invalidate the work as 'sustainable', so I would recommend further analysis to ascertain which of the so called SM techniques or works can actually be considered sustainable.

In terms of format, some improvement in the construction of some sentences is required, particularly when listing or enumerating. Inadvertently the verb has been included in singular instead of plural, and some times the sentences appear unfinished.

Other than that, I think it would make a nice contribution in a relatively unexplored area.

Kind regards

Reviewer 3 Report

Please see the file attached

Round 2

Reviewer 3 Report

The authors have successfully revised their manuscript and i strongly believe that it is now appropriate for publication.